# An Exploration of *Listeria monocytogenes*, Its Influence on the UK Food Industry and Future Public Health Strategies

**DOI:** 10.3390/foods11101456

**Published:** 2022-05-17

**Authors:** Joshua Macleod, Michael L. Beeton, James Blaxland

**Affiliations:** 1Microbiology and Infection Research Group, School of Sport and Health Sciences, Cardiff Metropolitan University, Western Avenue, Cardiff CF5 2YB, UK; sm80868@cardiffmet.ac.uk (J.M.); mbeeton@cardiffmet.ac.uk (M.L.B.); 2ZERO2FIVE Food Industry Centre, Llandaff Campus, Cardiff Metropolitan University, Western Avenue, Cardiff CF5 2YB, UK

**Keywords:** *L. monocytogenes*, listeriosis, biocide, antibiotic resistance, public health, food industry

## Abstract

*Listeria monocytogenes* is a Gram-positive intracellular pathogen that can cause listeriosis, an invasive disease affecting pregnant women, neonates, the elderly, and immunocompromised individuals. Principally foodborne, the pathogen is transmitted typically through contaminated foods. As a result, food manufacturers exert considerable efforts to eliminate *L. monocytogenes* from foodstuffs and the environment through food processing and disinfection. However, *L. monocytogenes* demonstrates a range of environmental stress tolerances, resulting in persistent colonies that act as reservoirs for the reintroduction of *L. monocytogenes* to food contact surfaces and food. Novel technologies for the rapid detection of *L. monocytogenes* and disinfection of food manufacturing industries have been developed to overcome these obstacles to minimise the risk of outbreaks and sporadic cases of listeriosis. This review is aimed at exploring *L. monocytogenes* in the UK, providing a summary of outbreaks, current routine microbiological testing and the increasing awareness of biocide tolerances. Recommendations for future research in the UK are made, pertaining to expanding the understanding of *L. monocytogenes* dissemination in the UK food industry and the continuation of novel technological developments for disinfection of food and the food manufacturing environment.

## 1. Introduction

*Listeria monocytogenes* is a Gram-positive, intracellular bacterial species recognised primarily as an opportunistic foodborne pathogen. The causative agent of listeriosis, *L. monocytogenes*, is responsible for life-threatening infections that arise typically in pregnant women, neonates, the elderly, and immunocompromised individuals. Resulting infections, that can present as gastroenteritis, septicaemia, and meningoencephalitis among others, have a hospitalisation rate of >90% and a mortality rate among non-pregnant individuals of between 15 and 25% in the UK [1,2].

There were 142 cases of listeriosis identified in England and Wales during 2019 [2]. Of these, 17.6% related to pregnant individuals, of which 63% resulted in stillbirth or miscarriage. Of the remaining, non-pregnancy related cases, the mortality rate was 19.7% [2]. Of the listeriosis outbreaks identified in England and Wales (where two or more cases were linked) between 2017 and 2019, all published investigations highlighted a link to food or food environment [2,3,4]. This not only results in a considerable impact on public health, but places considerable stress on the food industry as a whole to recognise and respond effectively to the pertinence of *L. monocytogenes* and its environmental stress and biocide tolerances.

One of the main hurdles faced by food manufacturers is the ubiquitous and environmentally tolerant attributes of *L. monocytogenes*, from which results a constant concern. Able to survive and grow between 0 and 45 °C, the pathogen can subsist in refrigerated areas designed to minimise the growth of microorganisms, resulting in *L. monocytogenes* reservoirs within production environments [5,6,7]. Other preservation techniques employed in food manufacturing are also tolerated to varying extents by *L. monocytogenes*, including osmotic tolerance to salt curing [7], pH tolerance to fermentation [8] and resistance to biocides including quaternary ammonium compounds (QAC) [9,10]. While responding to environmental stress, *L. monocytogenes* can procure cross-protection primarily through the employment of Sigma factor σB, among other alternative sigma factors (σC, σH, and σL), a transcription initiation factor that controls the general stress response of *Listeria* spp. and other Gram-positive bacteria, by the two-component regulatory systems [5,6,11].

As a means of overcoming these hurdles, food manufacturers introduce and develop food safety management systems (FSMS) designed around the food or raw ingredients produced or handled, facilities and instruments, and FSMS frameworks (such as Hazard Analysis and Critical Control Point (HACCP) or ISO 22000:2018) to demonstrate a combative effort against allergens, microorganisms, and chemical contaminants. Adherence to these systems has been the focus of studies in Europe, highlighting incidences whereby non-compliance to disinfection procedures has resulted in persistent populations [12].

In order to characterise *L. monocytogenes*, four distinct lineages have been described [13]. These lineages demonstrate varied distribution in environments, resulting in lineages that are overrepresented in, and therefore more pertinent to, food manufacturers [13]. Within these lineages, serotypes are described that, paralleling the lineages they reside within, are overrepresented in clinical and food manufacturing environments (FMEs) in the UK [14,15,16]. Increasingly, techniques such as multilocus sequence typing (MLST) are being employed for their greater discriminatory power, enabling *L. monocytogenes* to be further characterised into clonal complexes (CCs) that can help differentiate phenotypically distinct strains that may, for instance, harbour stress tolerances to food processing techniques, through sequence relatedness of housekeeping genes [17,18,19]. This can therefore assist food manufacturers in making alterations to FSMSs and reinterpret HACCPs if necessary.

The purpose of this review is to provide a comprehensive summary of outbreaks of *L. monocytogenes* in the UK as a foodborne pathogen, the logistical obstacles faced by food manufacturers in terms of stress and biocide resistance, and potential avenues for future research, surveillance and public health strategies.

## 2. Clinical Manifestations of Listeriosis and UK Outbreaks

Human listeriosis typically presents in one of three clinical syndromes: maternofoetal/neonatal listeriosis; bacteraemia with potential cerebral infections including meningoencephalitis, rhombencephalitis and encephalitis; and the predominantly self-limiting febrile gastroenteritis [20,21]. Focal infections including cutaneous listeriosis [22], endocarditis [23], septic arthritis and prosthetic joint infections [24], albeit less common, have also been described in the literature.

The infectious dose and dose–response relationship of *L. monocytogenes* have been explored in several studies [25,26], with Pouilett et al. [27] estimating a median infectious dose of 8.2 × 10^3^ in Listeriosis cases attributed to an outbreak in a dairy factory. Additionally, the incubation period of *L. monocytogenes* infection can vary broadly from 1 to 70 days in symptomatic cases [28,29]. This variance can be attributed to both the infectious dose and the time taken for the infection to colonise the site from which symptoms arise [25]. For non-invasive febrile gastroenteritis, symptoms arise after 12–48 h [28]. For invasive infections, the incubation period for bacteraemia is between 1 and 12 days, CNS between 1 and 15 days and pregnancy related cases between 10 and 67 days [28,29].

In the UK, the National Institute for Health and Care Excellence (NICE) recommends amoxicillin (or ampicillin) and gentamicin for the treatment of meningitis caused by *L. monocytogenes* in adults and children [30,31]. Treatment recommendations during pregnancy are difficult to ascertain and while amoxicillin and ampicillin are generally considered safe, gentamicin remains a US Food and Drug Administration (FDA) category D therapeutic where there is evidence of teratogenic risk to the foetus [32]. The UK Teratology Information Service, however, indicates that gentamicin is the preferred choice if aminoglycosides are evidenced to be effective for treatment [33].

Cases of listeriosis in England and Wales are outlined in surveillance reports conducted by the Gastrointestinal Pathogens Unit (GPU) of Public Health England (PHE)/UK Health Security Agency (UKHSA), wherein reported cases are collated with food history data via questionnaires and whole-genome sequencing in order to identify the outbreak source [2]. From 2006 to 2019, between 135 and 226 cases were reported annually. An outbreak source therein is recognised if the same strain (within 5 SNPs) is identified in two or more clinical cases. In the 14 outbreaks reported by PHE between 2017 and 2019, only 7 had their contamination source identified [2,3,4]. The difficulty in identifying the contamination source may be due in part to the variance in incubation periods and analysing food histories of patients. However, it is imperative in order to remove the food from shelves, highlight potential discrepancies in food safety and ensure the safety of at-risk groups [20].

While surveillance of human listeriosis in England and Wales has been undertaken since the 1980s, less than half of those with confirmed foodborne origins are further explored in peer-reviewed literature [15]. Consequently, patterns pertaining to food vehicles, likely contamination events and geographical dissemination can be missed. However, those that are outlined in the literature over the past 30 years highlight an increased understanding and the ongoing nature of foodborne listeriosis in the UK.

In the late 1980s, several studies described *L. monocytogenes* outbreaks and sporadic cases in the UK relating to pâté, vegetables, soft cheese, and cooked chicken [34,35,36,37,38]. Clinical manifestations in these instances included maternofoetal listeriosis, spontaneous abortion and meningitis. Potential food vehicles were discerned through conducting food histories with patients or patient families, with subsequent phage typing of implicated foods either from the patient’s home [34,35] or retailers [36,37,39].

From 1990 to 2000, between 114 and 136 cases of listeriosis were reported per year [40]. Fewer studies explored these outbreaks, though Graham et al. [41] highlight a potential link between ten cases in the Northern and Yorkshire NHS area that occurred in 1999 to four case reports in the early 2000s associated with ready-to-eat (RTE) foods from hospitals supplied by the same caterer. Outside of the UK, reported outbreaks and sporadic cases were still reported in the literature in the US and Europe, with dairy and RTE foods remaining common food vehicles [42,43,44,45].

At the turn of the millennium, an increase in publications following outbreaks is apparent [15]. This has been maintained throughout the subsequent 20 years, reinforced with listeriosis becoming a notifiable disease in 2010 [46]. There were 12 reported outbreaks spanning 1999 to 2019 which were found to be of nosocomial foodborne origin, related primarily to pre-prepared sandwiches (and one incident also related to RTE salads). Two of these incidences (2007 and 2017) were related to single, sporadic cases [15,47]. The remaining ten outbreaks were linked to 2–9 cases of human listeriosis [41,48,49,50,51]. Community-acquired outbreaks and sporadic cases linking epidemiological evidence to food vehicles have also been frequent since 2000, but remain largely unpublished [15]. However, several of these outbreaks have been the focus of studies including incidences related to ox tongue, pork pies, crab meat and frozen sweetcorn [47,52,53,54,55].

## 3. *Listeria* in Food Manufacturing Environments

UK legislation on microbiological criteria for foodstuffs outlines limits for *L. monocytogenes* colony forming units per gram (cfu/g) in RTE foods [56]. RTE foods able to support the growth of *L. monocytogenes* are required to have non-detectable cfu/g in 25 g of food product before leaving the food business operator’s (FBO’s) control, with a maximum limit of 100 cfu/g at the end of the shelf-life of the product. RTE foods unable to support the growth of *L. monocytogenes* are granted a maximum limit of 100 cfu/g for the duration of the product’s shelf-life. However, RTE foods intended for consumption by infants or for special medical purposes are required to have non-detectable levels in 25 g of food product throughout shelf-life.

More broadly, UK legislative guidelines, alongside the Food Safety Act of 1990, require the implementation of a FSMS that oversees Good Hygienic Practices (GHP) and principles outlined in HACCP food plans as well as criteria for routine testing of microbial activity in foodstuffs [56,57]. Internationally recognised standards, such as the British Retail Consortium’s global food safety standard (BRC), the Safe Quality Food (SQF) and ISO 22000:2018, seek to support manufacturers and retailers in ensuring consumer safety through stringent FSMS implementation throughout the manufacturing processes [57].

Insufficient compliance to these criteria can be manufacturer, foodstuff, or contaminant specific, with the latter potentially indicating inadequate GHP for the removal of the contaminant [58]. Financial or infrastructural limitations of a growing small and mid-sized enterprise (SME) may mean that FSMSs are developed in house by employees with potentially limited knowledge of food safety and manufacturing [57]. Though studies have indicated that compliance to in-house systems did not differ when compared to certified systems [59], hazard identification and control have been shown to be reduced [60]. This non-compliance or insufficient food safety can increase the likelihood of *L. monocytogenes*, or other contaminants, persisting in the environment, subsequently increasing the risk of food contamination reaching the consumer, particularly in minimally processed foods [12].

### 3.1. Occurrence of Listeria from Primary Food Production to Retail

The ubiquitous distribution of *Listeria spp*. results in numerous events that can introduce *Listeria* to the food manufacturing environment (FME), be it via raw ingredients early in processing or via environmental contact such as on the hands and clothing of workers [61]. The mechanisms by which *Listeria* is introduced are the subject of several studies, exploring key contamination events that predispose facilities to the distribution of contaminated foods and highlighting food safety oversights that can enable them to persist on surfaces in the environment [61,62].

Muhterem-Uyar et al. [62] describes three contamination scenarios observed in a study of 12 meat and dairy processing facilities in Europe. Sporadic contamination denoted surfaces from which *L. monocytogenes* were detected once in the study, typically early in the processing chain, where internal and external environments are more likely to interact. Hotspot contamination, conversely, referred to surfaces with repeated *L. monocytogenes* detection that potentially represented persistent colonies employing stress tolerance mechanisms and biofilm formation [12]. Pre-existing contamination control strategies, such as disinfection of food contact surfaces, may consequently corral *Listeria* to hotspot contamination surfaces that are improperly cleaned, such as drains and floors, or difficult to decontaminate such as nozzles, hoses, and machines with obstructive designs [12,63]. Lastly, widely disseminated contamination referred to situations in which large areas of the facility produced positive results for detection of *L. monocytogenes*. In Muhterem-Uyar’s study, this coincided with an absence of critical control areas [62]. Whereas sporadic and hotspot contamination occurred more frequently on surfaces without direct food contact, wide dissemination showed an increased incidence of surface contamination with which food made contact [62]. Though persistence can result in facility environment to food cross-contamination, the same can be said for raw material to processing facility cross-contamination via the introduction of a new *Listeria* population or the transfer of a resident strain from one harbourage site to another [64].

### 3.2. Current Contamination Control Strategies

Currently implemented strategies designed to minimise contamination of food and facility in the UK revolve around several key sections: cleaning and disinfection, food processing, infrastructural design and training of personnel [65]. As previously described above, these will follow criteria and guidelines provided by UK legislation and FSMSs and will be tailored towards the specific food product, potential contaminants, and the target consumer.

Cleaning and disinfection programmes in food processing environments target contaminants present on food contact surfaces (FCSs), such as utensils, equipment, and countertops, and non-food contact surfaces (NFCSs), such as floors, boots, and drainage. Cleaning of these surfaces typically involves a two-stage process led by an initial detergent used to remove grease, food particulates and residue followed by a disinfectant to inactivate microorganisms. Disinfectants such as QACs (benzalkonium chloride and didecyldimethylammonium chloride), peroxy/peroxo acids (peracetic acid) and inorganic chlorine-, base- and acid-based compounds (sodium hypochlorite and sodium hydroxide) are frequently employed in the food industry, though concentration and exposure time are decided by the food business following the recommendation from the disinfectant manufacturer [12,58]. The concentrations to which manufacturers recommend consider the removal of microbes to be below a safe limit while maintaining the quality of the food product [66]. These are increasingly met with intrinsic or acquired tolerances by *L. monocytogenes*, particularly with biofilm forming, persistent colonies exposed to sublethal concentrations [67,68,69].

Processing methods, such as thermal processing and irradiation, can function to increase the edibility and palatability of foods, but can also serve to inactivate pathogenic or spoilage-related microorganisms. Thermal processing, such as pasteurisation, is largely effective against *L. monocytogenes* at temperatures above 50 °C. However, any subsequent contact with the food processing environment can reintroduce bacteria to the food product [70]. Non-thermal techniques such as pulsed electric fields (PEF) and high-pressure processing (HPP) are frequently used for the treatment of liquid foods and ready-to-eat foods, respectively [6,71]. However, evidence of strain-dependent tolerances has been demonstrated by *L. monocytogenes* for both techniques, setting a worrying precedent [6,72]. The use of aqueous and gaseous ozone has also been shown to have an inhibitory effect of *L. monocytogenes*. Although efficacious, the effect of ozone on food handlers, the surrounding materials and food products themselves means that a certain concentration and contact time are required in order to be safe and effective [73].

### 3.3. Microbiological Testing of Listeria spp. in the Food Industry

To assess the safety of contamination control strategies outlined in FSMSs and to ensure safety of produce, UK legislation requires FBOs to test food products for contamination as described previously [56]. Additionally, testing of at-risk hotspots within the FME is advised to maintain the preservation of HACCP and minimise the persistence of pathogens, including *L. monocytogenes*, in the environment [74]. The food safety act 1990 also stipulates that an authorised officer of an enforcement authority can sample from any food or contact material, highlighting the importance of these surfaces [75]. Samples are primarily sent for microbiological testing at independent, United Kingdom Accreditation Service (UKAS)-accredited or non-accredited laboratories.

In the absence of otherwise specified methodologies, UK legislation recommends adherence to standards outlined in the ISO. For *L. monocytogenes* and other *Listeria* spp., ISO includes guidelines for sampling, enrichment and identification.

The sampling of food samples is outlined in ISO 17728:2015, and requires the food to be portioned where applicable using a suitable implement and stored in either a box or tube, dependent on state, and transported at an appropriate temperature to the laboratory [76]. ISO 18593:2018 pertains to the methods of sampling environmental surfaces in FMEs [77]. For the detection of microorganisms, the standard advises a surface area of 0.1 m^2^ to 0.3 m^2^ to be swabbed with a neutraliser-moistened cloth or sponge (or swab stick for areas difficult to reach) during or after production or after a process of disinfection. To recover microbiological material for analysis, the swab, sponge or cloth is recommended to be homogenised in 9–10 mL, 90–100 mL or 225 mL, respectively [77]. ISO 11290-1:2017 and 11290-2:2017 describe detection (presence-absence testing) and enumeration techniques, respectively, that select for *L. monocytogenes* and other *Listeria* spp. from both food and environmental sources [78,79].

For detection, the preliminary enrichment process includes two stages, detection taking place after each stage, and employs half-Fraser and Fraser broth for the first and second stage, respectively. Fraser broth contains two selective components, acriflavine and nalidixic acid, in a non-essential supplement and a chromogenic component, esculin, in the base [80]. Acriflavine, a RNA synthesis inhibitor, targets Gram-positive bacteria but, despite elongating the lag phase, *Listeria* spp. continue to grow [81]. Nalidixic acid, conversely, is incorporated into Fraser broth as an DNA synthesis inhibitor of Gram-negative bacteria [81]. The chromogenic component, esculin, is hydrolysed by esculinase (β-D-glucosidase) of *Listeria* spp. The product, esculetin, reacts with ferric ions to produce black colouration [80]. ISO 11290-1:2017 advises preparing a tenfold dilution of homogenised food or environmental inoculum in half-Fraser broth, incubating the resulting enrichment for 25 h ± 1 at 30 °C [78]. Half-Fraser, including the supplement with half the concentration of acriflavine and nalidixic acid, is recommended as a first stage for recovery of stressed or damaged *Listeria* spp. while suppressing undesired microorganisms [78,79,82]. For a second enrichment, a 0.1 mL inoculum is transferred to 10 mL of Fraser broth, containing double the concentration of selective compounds, and incubated at 24 h ± 2 at 37 °C.

ISO 11290-1:2017 recommends Agar *Listeria* according to Ottaviani and Agosti (ALOA) and a second selective medium chosen at the laboratory’s discretion, such as Oxford agar, for the detection of *Listeria* spp. [78,83]. ALOA incorporates selective and differential supplements. The former incorporates nalidixic acid, polymyxin B, ceftazidime and amphotericin or cycloheximide [78,79]. The latter includes L-α-phosphatidylinositol, the cleavage of which indicates the presence of *L. monocytogenes* and *L. ivanovii*-specific phosphatidylinositol-specific phospholipase C (PI-PLC) with an opaque halo [84]. For detection, two ALOA and two of the second selective medium are inoculated from both enrichment stages and incubated at 37 °C for 48 ± 2 h for ALOA and according to manufacturer’s specifications for the second medium [78].

For enumeration, as described in ISO 11290-2:2017, half-Fraser or buffered peptone water are recommended as diluents made up to a 1:10 dilution [79]. From this, 0.1 mL of the suspension (and subsequent decimal dilutions of this) is inoculated onto ALOA medium, incubated for 24 ± 2 h at 37 °C twice, checking CFU between and after incubations.

Confirmatory tests for both detection and enumeration of *L. monocytogenes* include beta-haemolysis, L-Rhamnose and D-Xylose [78,79]. For the detection method, microscopy is also mandatory [78]. Optional tests include catalase, motility and the CAMP test [78,79].

The detection methodology outlined above, as visualised in Figure 1 below, can take upwards of five days to confirm, including mandatory confirmatory tests. This is not conducive to a prompt response to the presence of *L. monocytogenes*, with the food manufacturer potentially withholding food while awaiting results. Additionally, it is noted in ISO 11290-1:2017 that *L. monocytogenes* may be undetected if in the presence of *L. ivanovii* or *L. innocua* [78]. As a result, other methodologies and media designed for the detection of *L. monocytogenes* and *Listeria* spp. in foodstuffs and the environment have been developed and subject to comparison in the UK and internationally [85,86,87,88]. Greenwood et al. [87] compared ALOA to Oxford agar and Rapid L. mono agar in both enumeration and detection methods. During enumeration, Oxford agar identified four *Listeria* spp. isolates not detected by ALOA and Rapid L. mono agar. Further, ALOA and Rapid L. mono agar detected one and two *L. monocytogenes* isolates, respectively, that were undetected by the other media. For detection, differences in *L. monocytogenes* identification between the three agars were not found to be statistically significant after the second enrichment. However, detection of other *Listeria* spp. was found to be significantly reduced on Rapid L. mono agar when compared to both ALOA and Oxford agar. Additionally, while interpretation of the agar plates after both the primary and secondary enrichment resulted in detection of 26 *L. monocytogenes* isolates, detection of other *Listeria* spp. was increased after secondary enrichment. While largely non-pathogenic, *L. ivanovii* has previously been implicated in human listeriosis cases in the UK [89]. Park et al. [88] compared selective agars PALCAM, Modified Oxford (MOX) agar, CHROMagar *Listeria* and Brilliance *Listeria* agar (BLA) with a novel lecithin and levofloxacin (LL) medium. Therein, statistically significant differences in specificities were not observed between LL medium, CHROMagar *Listeria* and BLA. However, specificity was considerably better than MOX. Curiously, correct morphology of *L. monocytogenes* was only observed on all media after 48 h, with atypical morphology observed after 24 h.

## 4. Stress Tolerances in Food Manufacturing Environments

*Listeria monocytogenes* demonstrates extensive stress tolerances, accompanied by the ability to replicate in a range of adverse conditions. As a result, the stresses encountered in FMEs resulting from food processing, disinfection or refrigeration may be met with persistence and recolonisation [12,90]. The extent of these tolerances and their ability to undermine food processing techniques have been the subject of several reviews [6,11]. A highly heterogeneous species, stress tolerances and, curiously, virulence have been demonstrated to differentiate when comparing lineages, serotypes, and clonal complexes [13,91,92,93]. This differentiation, and the recognition of lineage-specific accessory genes, may explain in some measure the preponderance of certain lineages and serotypes in different environmental niches [13,19].

As described briefly in the introduction, *L. monocytogenes* can be divided into four distinct lineages, with minimally overlapping serotypes described therein [13]. Lineage I, harbouring serotypes 1/2b, 3b and 4b, is well represented in both clinical and food isolates, though overrepresented in human cases. Lineage II, harbouring 1/2a, 1/2c and 3c, is overrepresented in food isolates. However, alongside 4b, 1/2a is among the highest implicated serotypes in human listeriosis in the UK and Europe [14,15]. Lineage III and IV are associated primarily with ruminants, though rarely isolated, with serotypes 4a, 4b and 4c described under both [13,16]. Further, MLST is conducted on seven housekeeping genes, resulting in combinatory sequence types (STs) which can be further clustered into CCs that relate to shared common ancestors [18,19]. Isolates within a CC typically, though not concretely, share a serotype or a narrow set of serotypes and therefore usually fall under one lineage [18]. For instance, CC1 and CC2 have been found to primarily fall under the clinically relevant serotype 4b, though isolates have also been characterised into serotypes 4d and 4e [18]. These classification schemes can allow for the monitoring of stress tolerance, that may be overrepresented in certain serotypes, throughout strains.

### 4.1. Biocides in Food Manufacturing

As described above, biocides are employed in FMEs in cleaning and disinfection procedures as part of routine food safety. Disinfectants such as QACs are frequently employed, aiming to eliminate new or persistent contaminants, but present the risk of selecting for resistant phenotypes through sublethal disinfection [12]. In 2002, a study in the UK measuring the necessary concentration of QAC and sodium hypochlorite to produce a 5-log reduction in *L. monocytogenes* found that concentrations 10 to 100-fold lower than recommended were sufficient [94]. However, more recent studies outside the UK have identified persistent colonies that remain after disinfection in the FME and explored mechanisms underpinning this resistance [12,95,96]. The preponderance of mechanisms currently understood to increase tolerance to biocides are intrinsic, including overexpression of efflux pumps, changes in membrane composition and the formation of biofilms, though adaptive tolerances have been described in the literature [97].

Conficoni and colleagues [12] explored the persistence of *L. monocytogenes* in meat-processing environments in Italy before and after exposure to the manufacturer’s disinfection strategy. Though it was highlighted that four of five manufacturers used incorrect disinfectant concentrations, several isolates were shown to contain genes encoding efflux pumps Lde and MdrL associated with biocide resistance [12]. Other efflux systems have been explored for their role in biocide resistance, including the *cadAC* cassette [98,99]. Containing a P-type ATPase-encoding gene, *cadA*, recognised for conveying cadmium resistance and *cadC*, a repressor of *cadA*, the cassette has been identified in benzalkonium chloride (BC) tolerant isolates [98,100]. Curiously, a plasmid (pLM80) known for harbouring *cadAC* in *L. monocytogenes* H7858 also carries the *bcrABC* cassette, consisting of a transcription regulator gene and two small multidrug resistance protein genes, that has been attributed to BC tolerance [96,98]. Another study observed the induction of efflux pump systems in previously biocide-susceptible *L. monocytogenes* isolates when exposed to an increasing concentration of BC [101]. Curiously, already resistant *L. monocytogenes* isolates that were further adapted did not demonstrate an induction of efflux pumps but instead showed alterations in cell membrane fatty-acid composition. Change in membrane composition occurs in response to other environmental stresses and prevents entry of foreign molecules [6]. Further, Romanova et al. [102] found that the incorporation of an efflux pump inhibitor, reserpine, did not reduce the minimum inhibitory concentration of BC in biocide resistant *L. monocytogenes* isolates. This suggests that efflux pumps may play little to no role in biocide resistance or may instead be one component of a larger stress response repertoire.

Biofilm has previously been implicated in *L. monocytogenes* survival in FMEs, enhancing tolerances to biocidal agents [103]. When measuring the concentration at which 90% of bacteria would be killed, or 90% lethal dose (LD90), of QAC against 5-day-old *L. monocytogenes* biofilms versus planktonic cells, Andrade et al. [104] found a statistically significant increase in tolerances to QACs in the biofilm group. Curiously, to achieve an LD90 in *L. monocytogenes* biofilms, a range of 298–532.2 ppm of QAC was required despite the manufacturers recommended a concentration of 150 ppm [104]. This suggests that even sufficient use of disinfectants according to manufacturer’s recommendations may be met by resistance, encouraging the persistence of tolerant strains.

*L. monocytogenes* can also enter a viable but non-culturable (VBNC) state, whereby cells maintain a reduced metabolic activity but no longer divide and undergo several physiological changes [105]. VBNC state has been induced in *L. monocytogenes* when exposed to suboptimal pH, desiccation, high temperature and chlorine-based biocides (BC) among others [106,107]. The potential for *L. monocytogenes* to retain virulence and potentially remain pathogenic while in the VBNC state has been explored [108]. Zolfaghari et al. [108] identified a continued expression of virulence factor genes *hly* and *inlA* after 27 days in VBNC *L. monocytogenes*. Despite this, *L. monocytogenes* in VBNC states were not resuscitated after being inoculated into mouse livers or spleens [109]. Further, VBNC *L. monocytogenes* inoculated into a human colon cancer cell line, HT-29, was deemed avirulent and unable to adhere to the cell line [110]. Despite this, VBNC *L. monocytogenes* has been demonstrated to be culturable once reintroduced to food products, indicating that a return to nutrient-rich and environmentally optimal conditions may support resuscitation [111].

### 4.2. Antibiotic Resistance and Biocide Cross-Resistance

*Listeria monocytogenes* demonstrates intrinsic tolerances to extended-spectrum cephalosporins, fosfomycin and fusidic acid [112]. Conversely, *L. monocytogenes* is naturally susceptible to a range of antibiotic classes including aminoglycosides, macrolides and penicillins [112]. This has understandably directed current therapeutic strategies. However, concerns for both acquired resistance and cross-resistance in *L. monocytogenes* in clinical, food and environmental samples are evident in the literature, including antibiotics currently employed in the UK for the treatment of listeriosis, placing pressure on these current therapeutic options [113,114,115]. Amoxicillin and ampicillin resistance have been identified in several studies, though very few of tested strains [114,115]. Gentamicin resistance has also been identified in *L. monocytogenes*, attributed to multidrug resistance phenotypes, with between 3 and 10 coexisting resistances [115]. Curiously, ampicillin and gentamicin resistance did not co-occur in this study; however, only one incident of ampicillin resistance was identified [115]. Further, gentamicin resistance has been observed in a single *L. monocytogenes* isolate within Ireland, though whether this resistance co-existed with resistance to amoxicillin was not stipulated [116]. Research into the acquired mechanisms underlying these resistances is limited, particularly with gentamicin. However, acquired genes have been identified, with Enterococci and Streptococci representing the primary sources, typically through conjugation [117]. Genes acquired through these means by *L. monocytogenes* target resistance to tetracycline, fosfomycin, fluoroquinolones, macrolides, penicillins and other β-lactams [117,118].

It has been considered, given the prevalence of antibiotic-resistant *L. monocytogenes* detected in food products, that resistance to these antibiotics may be employed via either other bacterial contaminants or the employment of mechanisms used to counteract disinfection in the manufacturing environment [96]. Efflux pumps, as described above, have been evidenced to support the removal of toxic substances such as biocides. However, antimicrobial agents may also be removed by efflux pumps, indicated by studies linking Lde in the removal of ciprofloxacin and other fluoroquinolones [119,120]. Jiang et al. [120] also identified cross-resistance between induced ciprofloxacin-resistant *L. monocytogenes* and ethidium bromide, with increased expression of Lde in four of the induced strains. Further, Rakic-Martinez et al. [121] found that selection by ciprofloxacin or ethidium bromide increased tolerances to these and to gentamicin. Exposure to reserpine, the efflux inhibitor, partially reduced these tolerances in *L. monocytogenes*, further suggesting the potential for efflux-pump mediated cross-resistance. Additionally, an efflux pump gene, *emrC*, has been shown to confer resistance to BC, amoxicillin and gentamicin [122]. However, another study that tested isolates from food manufacturers in Germany was unable to find the correlation between BC and antibiotics resistance, indicating that the mechanisms underlying BC resistance in these isolates, including *emrC*, did not confer cross-resistance to antibiotics, including penicillins and gentamicin [96].

## 5. Recommendations and Future Strategies

### 5.1. Research-Led Surveillance

The significance and capacity of public health–academic relations have been at the forefront of managing and understanding the COVID-19 pandemic. They have highlighted the pertinence of cooperative scientific enquiry at both the national and international scales. To that end, the value of cooperation not just for emerging infectious disease but well-established infectious disease cannot be understated.

With *L. monocytogenes*, several collaborative opportunities between public health, food manufacturers and academia can be considered. Primarily, public health surveillance of *L. monocytogenes* can be explored further by academic research to the benefit of both public health and food manufacturers. Current listeriosis surveillance conducted by UKHSA (formerly PHE) provides epidemiological data on isolates attributed to cases throughout England and Wales, providing insight into patient demographics, relatedness of isolates to food manufacturers or other known outbreaks, and the dissemination of the outbreak [2,3,4,15]. Research developing on these findings could explore the virulence capabilities of isolates, stress tolerances, and conduct meta-analyses of these attributes for all reported *L. monocytogenes* isolates. Results of the could tailor the implementation of symptomatic treatment strategies in patients, for instance. Additionally, identification of stress tolerances in an outbreak-related clonal complex can inform food manufacturers implicated in an outbreak as to where changes may be necessary.

Further, while clinically relevant *L. monocytogenes* isolates are identified through surveillance, an understanding of the epidemiological landscape of serotypes and clonal complexes in food manufacturers in the UK could further our understanding of persistence and likely routes of entry into both the manufacturing environment and food stuffs. The relationship between serotypes and tolerances to environmental stresses such as disinfection would further inform food manufacturers the most effective disinfection strategies. Lastly, with consumers showing an increased interest in minimally processed foods, the development of novel techniques for disinfection and processing can be adapted with an understanding of the Listerial landscape in FMEs in the UK.

### 5.2. Methodologies for Culturing Listeria spp.

As explored within this review, culturing techniques for routine surveillance of *L. monocytogenes* in the UK currently employ a two-stage detection process. This, along with confirmatory tests, can result in a 5 day window between sample collection and detection, leaving food manufacturers unable to distribute or potentially recall at-risk foods for that period. In order to reduce this, novel culturing methods and the employment of high-throughput molecular techniques could be employed. Unfortunately, many of these techniques can be cost prohibitive for both independent routine laboratories and food manufacturers. Additionally, *L. monocytogenes* is one of several pathogenic or spoilage-causing microorganisms that may need to be accounted for. Development of techniques that can amalgamate the detection of multiple microorganisms could be a time-saving and cost-effective endeavour for both laboratory and manufacturer. These techniques would also be required to equal or exceed current techniques approved in ISO standards in sensitivity and specificity. Other reviews have explored these alternatives with excellent depth [80,123]. Molecular techniques such as multiplex PCR could be optimised in several ways, thus overcoming certain constraints. Multiplex PCR, in which multiple genes are targeted for amplification, can be designed to target pertinent foodborne pathogens allowing for the amalgamation of several tests [124]. Consequently, *L. monocytogenes* and *Listeria* spp. can be differentiated, as originally developed by Ryu et al. [125], by targeting species-specific gene sequences, enabling for quick characterisation for the food manufacturer. Genes encoding *L. monocytogenes*-specific virulence factors, such as *actA*, *hlyA*, *iap* and *inlA,* have therefore been the target of multiplex PCR assays [123]. Similar simultaneous detection has been attainable for *L. monocytogenes* and other foodborne pathogens with DNA microarrays, using 16S rRNA and other target genes [126]. Non-molecular, immunological techniques such as lateral flow immunoassays have also been developed for the detection of *Listeria* spp. [127]. While this immunoassay was not specific to *L. monocytogenes*, it was able to produce positive results in the presence of *L. monocytogenes* in 15 min. More recently, a chitosan-cellulose nanocrystal (CNC) membrane-based fluorescence immunoassay was developed for the detection of *L. monocytogenes,* targeting a *L. monocytogenes*-specific region of the cell wall protein p60, PepD [128]. The sandwich enzyme-linked immunosorbent assay relied on anti-pepD capture monoclonal antibodies and anti-p60 fluorescently tagged polyclonal antibodies. The assay was able to detect *L. monocytogenes* within 8 h of enrichment and with concentrations of 10^2^ CFU/mL [128].

Additionally, with evidence of VBNC *L. monocytogenes* in food manufacturing environments, current detection methods using culture media may miss positive samples that are otherwise recoverable in food stuffs [106,107]. Molecular techniques such as quantitative PCR (qPCR) would overcome the initial hurdle of not requiring a culturable microorganism. However, this technique does not discriminate against DNA from inactive *L. monocytogenes* cells and therefore can result in false positives despite, for instance, an adequate disinfection regiment [128]. Thus, techniques such as PMA-stained qPCR may be incorporated. Containing a DNA-binding agent, propidium monoazide (PMA), the technique employs the agent’s ability to penetrate damaged cell membranes and bind irreversibly to DNA when exposed to light [129]. When bound, PMA prevents amplification. However, studies have suggested that PMA may not be activated effectively if in high microbial concentrations and may inhibit DNA amplification in instances of high levels of heat-killed *L. monocytogenes* alongside low levels of viable cells [130]. Another technique employs carboxy-fluorescein diacetate (CFDA), a fluorogenic ester, that is cleaved into the fluorescent product, carboxyfluorescein, by bacterial esterases of viable and VBNC cells [128]. The incorporation of these techniques could be considered in instances of suspected false negatives or if manufacturers are undergoing changes in disinfection strategy.

### 5.3. Future Research

While *L. monocytogenes* is well established in the literature, particularly pertaining to its impact on food manufacturing and public health, there is a limited repertoire of publications relating to *L. monocytogenes* in the UK. Numerous studies have been conducted throughout Europe that explore the distribution of serotypes in FMEs and have begun painting a picture of their relationships with virulence, stress tolerances and persistence [12,64]. This helps the food industry prepare for eventualities in which tolerant or persistent serotypes are identified, be this through changes in FSMSs or strengthening of disinfection strategy. In the UK, the current *L. monocytogenes* serotype distribution is inferred through clinical isolates obtained from sporadic cases or outbreaks [15,47]. However, given the propensity of serotypes under lineage I to be identified in human isolates, this method may underrepresent the remaining population observed in the FME [13]. Those underrepresented through the current reporting method are, given their higher diversity and recombination levels, likely to present a greater array of tolerances to adverse environments [6,13]. For these reasons, an exploration of *L. monocytogenes* in the UK food industry will help elucidate the international and national distribution of a highly diverse species, benefiting both public health and the food manufacturing industry.

Furthering this, as has been explored in this review, stress tolerances expressed by *L. monocytogenes* are increasingly acknowledged in the literature [6]. While the mechanisms underpinning these tolerances are still investigated, the development of novel food processing and disinfection techniques may help curtail the persistence of *L. monocytogenes* in particularly at-risk environments. Morey et al. [131], for instance, explored the use of UV light against a cocktail of *L. monocytogenes* strains on conveyor belts and found significant reduction in CFU to below detectable limits.

The use of bacteriophages has also been explored as a potential *L. monocytogenes* growth inhibitor, with PhageGuard Listex™ (formerly Listex™ P100) at the forefront of current research [132]. With these bacteriophages, after entering the cell and producing progenies, lytic enzymes kill the host cell, thus releasing the phage back into the environment [132]. Though efficacious, the ability of the technology to reduce *L. monocytogenes* to undetectable levels depends on food matrix, exposure time, pH and the subsequent storage of the food product [132]. Additional concerns have emerged after tolerances to the bacteriophage were observed in *L. monocytogenes*, perhaps due to alterations to the phage receptor or through the CRISPR-Cas system, in below 5% of isolates [133].

Recently, ozone has been considered as a disinfectant against *L. monocytogenes* and its biofilm in FMEs given its lower environmental impact and ability to diffuse through obstructive machinery into potential hotspots [73,134]. The application of ozone has been considered for treatment of food sources and the processing environment. Panebianco et al. [134] tested the effect of ozone at 50 ppm on *L. monocytogenes* in planktonic and biofilm states. Planktonic cells were completely inactivated within 6 h. However, only 59% of biofilm formers demonstrated a significant reduction in biomass [134] Conversely, another study found that exposure to 4 ppm completely eliminated *L. monocytogenes* in biofilm though this was a 16-fold greater than the 0.25 ppm required to eliminate planktonic cells [103]. Curiously, a study of *E. coli* and *L. monocytogenes* exposed to gaseous ozone found that biofilm, quorum sensing and components of the two-component regulatory system were altered and, in some instances, upregulated, though not in a time-dependent manner [135]. The results of these studies indicate firstly that a further exploration of ozone and biofilm interactions is necessary; and secondly, if additional ozone concentrations would be required to eliminate biofilms in foods or the processing environment, the effect of ozone on taste, visual and nutritional quality would be required. What may prove practical is the combined use of these technologies with one another or with traditional disinfection strategies to minimise persistence or remove *L. monocytogenes* from foodstuffs.

Lastly, the role that FMEs play in antibiotic resistance is worthy of exploration, particularly given the detection of antibiotic-resistant *L. monocytogenes* in foodstuffs [113,114,115]. The role of efflux pumps is contentious, though several have tentatively been implicated in resistances to gentamicin, amoxicillin, fluoroquinolones, such as ciprofloxacin [96,119,120,121]. Given the recommendations to use amoxicillin and gentamicin for the treatment of listeriosis, confirmation of these mechanisms and the potential for them to be induced under particular conditions are crucial.

## 6. Conclusions

*L. monocytogenes* is a demonstrable concern for public health and the food manufacturing industry. With a sizeable history of outbreaks in the UK, it has been a pressing issue for decades with minimal UK-centric research monitoring and a burden to public health and those affected by listeriosis. Current methodologies employed to routinely identify *L. monocytogenes* are of a very high standard, ensuring where possible the safety of the public. However, these techniques can result in a 5 day window between sample dispatch and results. Further, current disinfection and HACCP strategies have been met with persistent colonisation by *L. monocytogenes*, suggesting an insufficient employment of effective disinfection techniques and intrinsic and acquired tolerances to the current use of biocides.

The application of novel technologies requires continuation of research into their efficacy for the elimination of *L. monocytogenes* and other foodborne pathogens in FMEs and their respective foodstuffs. Given the limited research undergone in the UK, a bilateral approach may elucidate the role that *L. monocytogenes* is going to play in the food manufacturing industry moving forward—firstly, through an investigation into the serotype distribution of *L. monocytogenes* in the UK food industry and its relatedness to clinical isolates, environmental stress tolerances and antibiotic resistance; secondly, through an investigation into the employment of novel technologies for microbial disinfection and its efficacy against *L. monocytogenes* of varying phenotypes.

## Figures and Tables

**Figure 1 foods-11-01456-f001:**
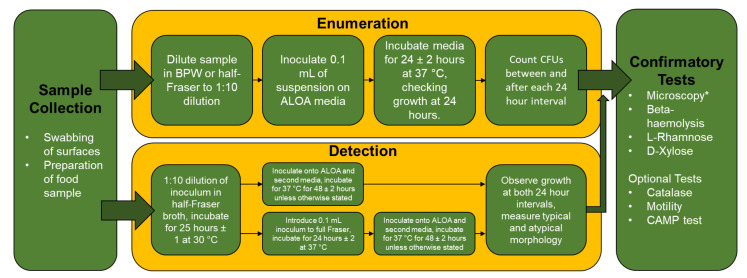
A workflow chart of the ISO approved methods for the culturing and enumeration or detection of *L. monocytogenes* from food or environmental samples. BPW: buffered peptone water; ALOA: Agar Listeria according to Ottaviani and Agosti. * Microscopy is mandatory for the detection method and optional for enumeration [76,77,78,79].

## Data Availability

Data is contained within the article or supplementary material.

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
