# Peer review of "An Exploration of Listeria monocytogenes, Its Influence on the UK Food Industry and Future Public Health Strategies"

_foods, 2022, doi:10.3390/foods11101456_

Round 1

Reviewer 1 Report

The present article focuses on the occurrence of Listeria monocytogenes in the UK, its basic characteristics and methods of detection, as well as the problems with Listeria in food manufacturing environments.

The manuscript is quite extensive, but some of its chapters contain only very brief information that would be good to expand and supplement. For example, line 34 shows mortality among non-pregnant individuals, and it would certainly be interesting to know the mortality of pregnant women.

Line 53 mentions the role of the Sigma factor σB in the bacterial response to stress conditions. However, it is a complex process involving a number of factors, so it would be a shame to limit the review to just one of them.

Lines 72-77 only very generally mention the use of MLST method to characterize Listeria monocytogenes strains. Please could you provide more detailed information about CCs of Listeria monocytogenes and their relationship to known serotypes?

In lines 91-97, the authors focus on the incubation period of L. monocytogenes infection in different types of diseases. However, these data do not correspond to the data in the article to which the authors refer [28].

Chapter 2.1 is interesting and contains well-organized information on UK-related epidemics. Only the last paragraph (lines 150-156) can be considered redundant, as can some paragraphs of subsequent chapters (lines 399-403 and 443-446) and possibly the whole chapter 5.1.

The authors described the culturing techniques in great detail and describe their time complexity. Unfortunately, new methods that can solve this problem are only described very briefly in this manuscript. It could be interesting to mention, for example, targeted genes in PCR and also targeted antigens in the immunochemical methods. For the latter, it would also be useful to mention formats other than just a lateral flow immunoassay, i.e. an ELISA method specific for L. monocytogenes or NALFIA.

Author Response

REVIEW 1

The present article focuses on the occurrence of Listeria monocytogenes in the UK, its basic characteristics and methods of detection, as well as the problems with Listeria in food manufacturing environments.

The manuscript is quite extensive, but some of its chapters contain only very brief information that would be good to expand and supplement. For example, line 34 shows mortality among non-pregnant individuals, and it would certainly be interesting to know the mortality of pregnant women.

Unfortunately, data from PHE does not include mortality rate among pregnant women. Changes have been made to miscarriage or stillbirth data to more accurately reflect the risk of materno-foetal listeriosis.

Line 53 mentions the role of the Sigma factor σB in the bacterial response to stress conditions. However, it is a complex process involving a number of factors, so it would be a shame to limit the review to just one of them.

Thank you for this, other sigma factors with known roles in the listerial stressosome have been included.

Lines 72-77 only very generally mention the use of MLST method to characterize Listeria monocytogenes strains. Please could you provide more detailed information about CCs of Listeria monocytogenes and their relationship to known serotypes?

Several changes have been made to this section. In the introduction, mention of MLST is more broad now to fit the rest of the introduction. Another paragraph has been added to section 4 to fit the flow of the paper and MLST and CC/serotype association is explored more.

In lines 91-97, the authors focus on the incubation period of L. monocytogenes infection in different types of diseases. However, these data do not correspond to the data in the article to which the authors refer [28].

Thank you for pointing this out. Data points now reflect that of the two references.

Chapter 2.1 is interesting and contains well-organized information on UK-related epidemics. Only the last paragraph (lines 150-156) can be considered redundant, as can some paragraphs of subsequent chapters (lines 399-403 and 443-446) and possibly the whole chapter 5.1.

Thank you for highlighting these. We agree that these sections did not add much substance to the review and have, excluding section 5.1, been removed. We did feel that with the removal of the other sections, that section 5.1 now has a better flow within the document.

The authors described the culturing techniques in great detail and describe their time complexity. Unfortunately, new methods that can solve this problem are only described very briefly in this manuscript. It could be interesting to mention, for example, targeted genes in PCR and also targeted antigens in the immunochemical methods. For the latter, it would also be useful to mention formats other than just a lateral flow immunoassay, i.e. an ELISA method specific for L. monocytogenes or NALFIA.

Absolutely, target genes of PCR have been highlighted and a more novel ELISA model has been included which is a nice addition to the review – thank you.

Reviewer 2 Report

This is a well written manuscript that flows logically. The minor comments are:

Line 72: Full-stop and space required at the end of the sentence.

Line 78: It may be more appropriate to use the words 'summary of outbreaks' as indicated in the Abstract instead of the word incidences, which may confuse the reader because the term incidences refers to new cases.

Line 3: ?superscript

Line 293, 310 and 323: Use italics for L. ivanovii

Line 386: the metablism for VNBC is at low rate

Author Response

Comments and Suggestions for Authors

This is a well written manuscript that flows logically. The minor comments are:

Line 72: Full-stop and space required at the end of the sentence.

            Thank you for highlighting this, it has been correct.

Line 78: It may be more appropriate to use the words 'summary of outbreaks' as indicated in the Abstract instead of the word incidences, which may confuse the reader because the term incidences refers to new cases.

            Quite right – this has been changed accordingly

Line 3: ?superscript

            We could not see what this was in relation too.

Line 293, 310 and 323: Use italics for L. ivanovii

            These have all been corrected – thank you.

Line 386: the metablism for VNBC is at low rate

              Thank you, this has been amended.

Reviewer 3 Report

This review connects current acknowledges on Listeria monocytogenes in the field of research and management strategies, especially for the UK. The manuscript is generally well written and would be of interest to a substantial portion of food microbiologists and risk managers. I have relatively minor comments only.

General. The description on Listeria types in section 1 is unexpected (although it is very important), or it is insufficient to the whole manuscript. I suggest authors to better review and explain Listeria types in food and epidemiology, introduce more details and research progresses on the relation between Listeria types and stress tolerances.

General. It is unreasonable to set 2.1 as an individual part. Please optimise the skeleton and reorder the writing of section 2.

General. It is better to present and compare details of different Listeria (qualitative and quantitative) testing methods with a readable table in section 3.

Detail 1 as follows:

Line 90 delete “.” after [27]
Line 91 … of 8.2×103 in listeriosis …
Line 111 From 2006 to 2019, …
Line 115 … analysing the food safety …
Line 117 … ensure the safety of …
Line 145 … linked epidemiological evidence to …
Line 152 sources
Line 168 Good Hygienic Practices (GHP)
Line 170 … the British Retail …
Line 171 the Safe Quality Food (SQF) program, and ISO …
Line 176-177 Small and Mid-size Enterprise (SME)
Line 189 change the second “is” to “are”
Line 194-196 Unclear, please rephrase.
Line 220-221 countertops
Line 228 the recommendation
Line 230 change “below” to “be below” or delete it
Line 235 the edibility
Line 239 pulsed
Line 249 the safety
Line 251 the preservation … the persistence of …
Line 257 recommends
Line 261 to be portioned …
Line 278 change “an” to “a”
Line 292 do you mean “cycloheximide”?
Line 293 “L. ivanovii” should be italicized
Line 310 “L. ivanovii” and “L. innocua” should be italicized
Line 314, 315, 316, 320 change “L. mono” to “L. mono agar”
Line 317 deifferences
Line 319 the second enrichment
Line 384 the persistence
Line 378 90% lethal dose (LD90)
Line 397 nutrient-rich
Line 440 the correlation
Line 441 “emrC” should be italicized
Line 451 scales
Line 514 carboxy-fluorescein diacetate (CDFA)

Detail 2 as follows:

Line 46      Pay attention to the unit format. There should be a space between the number and the unit. For example, 45°C should be changed to 45 °C. The same below.

Line 64-77   There is a logical problem between this paragraph and the above. The first section introduces the characteristics and pathogenicity of L. monocytogenes. Then the food risk management is introduced. At this point, L. monocytogenes is suddenly returned to introduce its pedigree. Logic doesn't work.

Line 90      Pay attention to punctuation and scientific notation.

Line 95      The unit here is hours, and the unit below is h. Please unify the whole text.

Line 98-105  As mentioned above, there are three clinical syndromes of L. monocytogenes, but only the treatment methods for meningitis and mother-fetal neonates are described here, and the treatment methods or drugs for the predominantly self-limiting febrile gastroenteritis need to be supplemented.

Line 106     Only subtitle 2.1, this title can be deleted. This part introduces the relevant epidemic situation in the UK in detail according to the chronological order. It can also further clarify the corresponding pathogenic source (what kind of food), the pathogenic object (children or pregnant women or ordinary individuals) and the resulting disease (which of the three clinical syndromes) in the form of tables.

Line 147     Citation format should be noted.

Line 160     Pay attention to the unit format, there should be a space between the number and the unit. The same below

Line 268     ml should be changed to mL, the same below.

Line 330     Units please unify the whole text.

Line 550     The unit needs to be abbreviated and should be changed to min.

Line 605     There is an error in the format of the reference. Please check the journal requirements carefully. All citations have no DOI number.

Author Response

This review connects current acknowledges on Listeria monocytogenes in the field of research and management strategies, especially for the UK. The manuscript is generally well written and would be of interest to a substantial portion of food microbiologists and risk managers. I have relatively minor comments only.

General. The description on Listeria types in section 1 is unexpected (although it is very important), or it is insufficient to the whole manuscript. I suggest authors to better review and explain Listeria types in food and epidemiology, introduce more details and research progresses on the relation between Listeria types and stress tolerances.

Quite right, it did not fit the flow of the introduction so several changes have been made to rectify this. Firstly, the section in the introduction has been reduced in depth significantly whilst still mentioning it briefly given its pertinence. In order to retain the information therein, a paragraph has been added to section 4 just before 4.1 which is more indepth. Thank you for highlight this and for the recommendation.

General. It is unreasonable to set 2.1 as an individual part. Please optimise the skeleton and reorder the writing of section 2.

The subsection title has been removed.

General. It is better to present and compare details of different Listeria (qualitative and quantitative) testing methods with a readable table in section 3.

A diagram to explore current ISO culturing methodologies has been included to make the section more readable – thank you for the recommendation.

Detail 1 as follows:

Line 90 delete “.” after [27] Corrected
Line 91 … of 8.2×103 in listeriosis … Corrected
Line 111 From 2006 to 2019, … Corrected
Line 115 … analysing the food safety … We believe food history is more appropriate here
Line 117 … ensure the safety of …
Line 145 … linked epidemiological evidence to … Corrected
Line 152 sources Corrected
Line 168 Good Hygienic Practices (GHP) Corrected
Line 170 … the British Retail … Corrected
Line 171 the Safe Quality Food (SQF) program, and ISO … Corrected
Line 176-177 Small and Mid-size Enterprise (SME) Corrected
Line 189 change the second “is” to “are” Corrected
Line 194-196 Unclear, please rephrase. Corrected
Line 220-221 countertops Corrected
Line 228 the recommendation Corrected
Line 230 change “below” to “be below” or delete it Corrected
Line 235 the edibility Corrected
Line 239 pulsed Corrected
Line 249 the safety Corrected
Line 251 the preservation … the persistence of … Corrected
Line 257 recommends Corrected
Line 261 to be portioned … Corrected
Line 278 change “an” to “a” Corrected
Line 292 do you mean “cycloheximide”? Corrected
Line 293 “L. ivanovii” should be italicized Corrected
Line 310 “L. ivanovii” and “L. innocua” should be italicized Corrected
Line 314, 315, 316, 320 change “L. mono” to “L. mono agar” Corrected
Line 317 deifferences Corrected
Line 319 the second enrichment Corrected
Line 384 the persistence Corrected
Line 378 90% lethal dose (LD90) Corrected
Line 397 nutrient-rich Corrected
Line 440 the correlation Corrected
Line 441 “emrC” should be italicized Corrected
Line 451 scales Corrected
Line 514 carboxy-fluorescein diacetate (CDFA) Corrected

Thank you for highlighting these errors, changes have been made where applicable.

Detail 2 as follows:

Line 46      Pay attention to the unit format. There should be a space between the number and the unit. For example, 45°C should be changed to 45 °C. The same below.

These changes have been made throughout – thank you for highlighting them.

Line 64-77   There is a logical problem between this paragraph and the above. The first section introduces the characteristics and pathogenicity of L. monocytogenes. Then the food risk management is introduced. At this point, L. monocytogenes is suddenly returned to introduce its pedigree. Logic doesn't work.

As explained above – thank you again.

Line 90      Pay attention to punctuation and scientific notation.

Corrected

Line 95      The unit here is hours, and the unit below is h. Please unify the whole text.

These have now been corrected throughout the manuscript

Line 98-105  As mentioned above, there are three clinical syndromes of L. monocytogenes, but only the treatment methods for meningitis and mother-fetal neonates are described here, and the treatment methods or drugs for the predominantly self-limiting febrile gastroenteritis need to be supplemented.

We agree with this comment, to the best of our knowledge we could not find any specific UK based therapy that we could include here, so felt at this time it would be inappropriate to do so without a specific reference.

Line 106     Only subtitle 2.1, this title can be deleted. This part introduces the relevant epidemic situation in the UK in detail according to the chronological order. It can also further clarify the corresponding pathogenic source (what kind of food), the pathogenic object (children or pregnant women or ordinary individuals) and the resulting disease (which of the three clinical syndromes) in the form of tables.

Thank you for this suggestion – we attempted to construct tables but felt that they would be too extensive for inclusion in this review and was perhaps better communicated, after the other changes made from both reviewers, as text.

Line 147     Citation format should be noted.

Amended

Line 160     Pay attention to the unit format, there should be a space between the number and the unit. The same below

Corrected throughout

Line 268     ml should be changed to mL, the same below.

Corrected throughout

Line 330     Units please unify the whole text.

Corrected throughout

Line 550     The unit needs to be abbreviated and should be changed to min.

Corrected.

Line 605     There is an error in the format of the reference. Please check the journal requirements carefully. All citations have no DOI number.

We have checked the journal’s referencing requirements and made amendments where appropriate